# Emerging Non-Antibiotic Options Targeting Uropathogenic Mechanisms for Recurrent Uncomplicated Urinary Tract Infection

**DOI:** 10.3390/ijms24087055

**Published:** 2023-04-11

**Authors:** Yu-Chen Chen, Wei-Chia Lee, Yao-Chi Chuang

**Affiliations:** 1Graduate Institute of Clinical Medicine, College of Medicine, Kaohsiung Medical University, Kaohsiung 80756, Taiwan; 2Department of Urology, Kaohsiung Medical University Hospital, Kaohsiung Medical University, Kaohsiung 80756, Taiwan; 3Regenerative Medicine and Cell Therapy Research Center, Kaohsiung Medical University, Kaohsiung 80756, Taiwan; 4Division of Urology, Kaohsiung Chang Gung Memorial Hospital and Chang Gung University College of Medicine, Kaohsiung 833, Taiwan; 5Center for Shock Wave Medicine and Tissue Engineering, Kaohsiung Chang Gung Memorial Hospital, Kaohsiung 833, Taiwan

**Keywords:** recurrence, urinary bladder, urinary tract infections, women

## Abstract

Urinary tract infections (UTIs) are the most frequent bacterial infections in the clinical setting. Even without underlying anatomic or functional abnormalities, more than 40% of women experience at least one UTI in their lifetime, of which 30% develop recurrent UTIs (rUTIs) within 6 months. Conventional management with antibiotics for rUTIs may eventually lead to the development of multidrug-resistant uropathogens. Targeting of the pathogenicity of rUTIs, the evolution of uropathogenic *Escherichia coli* (UPEC), and inadequate host defenses by immune responses should be explored to provide non-antibiotic solutions for the management of rUTIs. The adaptive evolution of UPEC has been observed in several aspects, including colonization, attachment, invasion, and intracellular replication to invade the urothelium and survive intracellularly. Focusing on the antivirulence of UPEC and modulating the immunity of susceptible persons, researchers have provided potential alternative solutions in four categories: antiadhesive treatments (i.e., cranberries and D-mannose), immunomodulation therapies, vaccines, and prophylaxis with topical estrogen therapy and probiotics (e.g., *Lactobacillus* species). Combination therapies targeting multiple pathogenic mechanisms are expected to be a future trend in UTI management, although some of these treatment options have not been well established in terms of their long-term efficacy. Additional clinical trials are warranted to validate the therapeutic efficacy and durability of these techniques.

## 1. Introduction

Urinary tract infections (UTIs) are the most common bacterial infections in the outpatient setting and occur mainly in women [1]. More than 40% of women experience acute cystitis during their lifetime [2]. Among young women without underlying anatomical or functional abnormalities, 25% experience a recurrence of cystitis within 6 months of the initial UTI. The first-line treatment for uncomplicated UTIs depends on the use of a short-course antimicrobial regimen [3]. However, conventional prophylaxis and treatment of UTIs using antibiotics can result in the long-term alteration of the normal microbiota of the vagina and gastrointestinal tract and in the growth of multidrug-resistant (MDR) microorganisms. From a socioeconomic perspective, recurrent UTIs (rUTIs) may worsen global public health, lead to a high economic burden, and affect the quality of life of women [4].

rUTIs are defined as at least two symptomatic episodes within 6 months or at least three episodes within a period of 1 year. From the viewpoint of pathogenicity, the offending uropathogens and host susceptibility or a combination of both are responsible for the occurrence of rUTIs. Indeed, up to 80% of UTIs are caused by *Escherichia coli*, which accounts for half of the estimated global antibiotic resistance [5,6]. Therefore, rUTIs may present as either relapsing infections due to an incomplete clearance of uropathogens within two weeks of the treatment course or reinfection, which occurs two weeks after treatment completion. The majority of rUTIs are clinically categorized as reinfections [7]. On the other hand, women are vulnerable to UTIs because of their anatomical structure [8], including the fact that women have (1) a short distance between the urethral meatus and non-sterile anus, which allows for easier access for uropathogens to the lower urinary tract, and (2) a short urethral length without a complex structure, which facilitates the entry of uropathogens into the bladder. In addition, susceptibility to rUTIs may increase with frequent sexual intercourse and the use of spermicide in young women, in addition to increased prevalence in post-menopausal women due to their deficiency of estrogen, resulting in uropathogenic colonization [9].

Overall, the investigation of alternative strategies to decrease the occurrence of rUTIs is of interest to physicians and scientists. We believe that a better understanding of the pathogenicity of uropathogens, particularly for uropathogenic *Escherichia coli* (UPEC), can lead to the development of antivirulence treatment strategies other than antibiotics. Here, we review the components of the pathogenicity of uncomplicated rUTIs and virulence factors of UPEC and discuss potential approaches for targeting of the pathogenicity to limit rUTIs.

## 2. How Do Microorganisms Become Uropathogens?

Commensal microorganisms normally reside in the lower gastrointestinal tract of humans as a part of their microbiota. Few non-pathogenic organisms may acquire specific virulence through successful horizontal gene transfer, further causing UTIs [10,11]. Among these, UPEC is the main uropathogenic organism [12]. Other bacteria that cause uncomplicated UTIs include *Proteus mirabilis*, *Klebsiella pneumonia,* and other *Enterobacteriaceae*.

### 2.1. Uropathogenic Escherichia coli (UPEC)

*E. coli* strains that cause UTIs are termed UPEC. UPEC, which belongs to the group of extraintestinal *E. coli*, can be classified into eight serogroups (O1, O2, O4, O6, O7, O16, O18, and O75) based on its distinct surface O antigens, which represent the different functions of the variable repeating terminal polysaccharides attached to lipopolysaccharide [13], and is principally classified into two phylogenetic groups (B2 and D) based on a phylogenetic analysis using multilocus enzyme electrophoresis [14,15] (Table 1).

### 2.2. The Virulence Factors of UPEC

UPEC is different from harmless commensal fecal *E. coli* owing to the expression of particular virulence factors that specifically enhance their ability to colonize the urinary tract and cause UTIs [21]. Several key virulence factors include: (1) variable adhesins, as the crucial step for the irreversible attachment and adherence to uroepithelial cells over the lower urinary tract [22]; (2) the three main toxins secreted by UPEC: α-hemolysin, cytotoxic necrotizing factor 1 (CNF-1), and autotransporters, among which α-hemolysin is the most vital virulence factor associated with pyelonephritis. These toxins enhance exfoliation of the bladder epithelial cells, cause apoptosis of the host cells, cross mucosal carriers, and enable access to nutrients and iron stores of the host [23]; (3) autotransporters, consisting of secreted autotransporter toxin (SAT) and vacuolating autotransporter toxin (VAT), which induce cellular damage in renal and bladder cells, vacuole formation, cell elongation, autophagy induction, degradation of coagulation factor V, and dysregulation of the epithelial barrier of the bladder [24,25,26]; (4) systems for iron acquisition, namely siderophores, which ensure adequate levels of intracellular iron to maintain the survival of bacteria within the host urinary tract [22]; and (5) flagella, which mediate motility to enhance the ascending nature of UPEC [22]. Table 2 summarizes the various adhesive fimbriae, toxins, and autotransporters that contribute to virulence in UPEC.

## 3. Acute Uncomplicated Cystitis

Acute uncomplicated cystitis (AUC) occurs when uropathogens from the bowel or vagina colonize the periurethral mucosa and reach the bladder, causing a symptomatic UTI. Infections of the urinary tract depend on the combination of virulent invasion of uropathogens, inadequate host defenses by the immune response, and other predisposing factors.

### 3.1. The Adaptive Evolution of UPEC in UTI to Thrive within Urothelial Cells

The adaptive evolution of uropathogen infection includes two mechanisms: invasion of uroepithelial cells and persistence to survive intracellularly (Figure 1).

After the colonization of uropathogens, the preliminary step of UTI relies on adhesins, especially type I fimbriae, which enable uropathogens to recognize specific extracellular matrix proteins (laminin, fibronectin, collagen, and Tamm–Horsfall proteins) on the bladder epithelium, thereby binding to uroepithelial cells [27,28]. Furthermore, P fimbriae related to the process of invasion mediate other mannose-resistant adhesins. At this moment, some uropathogens are initially cleared by the first wave of host inflammatory responses, while others avoid the host’s defenses by intracellular invasion or morphological changes that are resistant to neutrophils. After intracellular invasion, they reside within vesicles by forming biofilms and undergo rapid and massive replication to establish intracellular bacterial communities within the cytosol of urothelial cells [22,41,42]. In addition, biofilms, which manifest as the presence of an extracellular matrix capable of containing proteins; polysaccharides; and DNA increase both the persistence and antibiotic resistance of intracellular UPEC.

However, the intracellular environment is inadequate for the survival of UPEC. Therefore, several strategies were developed for the intracellular survival of UPEC, which are summarized in Table 3, including (1) aerobic respiration to establish UTI by intracellular oxygen scavenging from the cytosol through cytochrome *bd* [43], the aim of which is to antagonize host apoptosis, thereby protecting intracellular uropathogens from the host cells’ exfoliation and supporting their replicative environment; (2) iron acquisition for cellular processes and bacterial growth by siderophores and/or other iron acquisition systems; the siderophores, known as yersiniabactin, salmochelin, enterobactin, and aerobactin, are capable of scavenging iron from the environment and further concentrating it in the bacterial cytosol. Different stains of UPEC have evolved iron acquisition systems to swipe iron from the host; furthermore, significantly higher levels of yersiniabactin and salmochelin were reported in UPEC than that in *E.coli* from the rectum, indicating their specific role in pathogenesis within the urinary tract [44]; (3) zinc acquisition by zinc-transport systems for iron hemostasis and relief of oxidative stress to maintain cellular function [45]; and (4) toxin secretion, which results in cytotoxic activity and destruction of the immune cells, further gaining access to nutrients from the hosts [14].

### 3.2. Host Defenses by Immune Response

Upon causing acute cystitis, UPEC induces several responses in the host against uropathogens in the bladder, including (1) inflammation by secreting cytokines, such as IL-1β, IL-6, and IL-8, to kill bacteria and to recruit the hosts’ inflammatory cells, namely neutrophils, monocytes, and mast cells, from the body to the uroepithelium [47]. The activation of Toll-like receptor 4 by the lipopolysaccharide, which is released by UPEC, increases the intracellular cAMP and causes the exocytosis of fusiform vesicles containing uropathogens [48,49]; (2) classical extrinsic and intrinsic apoptotic cascades of uroepithelial cells activated by FimH following the binding between type I fimbriae and uroplakins [50]. The exfoliation of superficial uroepithelial cells helps to eliminate intracellular uropathogens; furthermore, cell autophagy is critical in treatment to mediate the persistence of uropathogens in the bladder [51]; (3) antimicrobial defenses by upregulated antimicrobial peptides and proteins such as pentraxin 3, ribonuclease 7, cathelicidin, and defensins secreted by the urothelium and resident immune cells [52]; (4) deprivation of nutrients, such as by decreasing the amount of iron available to the pathogens, by the host’s innate immune system activated during UTI to limit the growth of uropathogens [53,54]. The host’s neutrophils produce lipocalin-2, which is capable of binding to the enterobactin siderophore against enterobactin-mediated iron uptake by UPEC. Notably, other UPEC evolved to utilize salmochelin siderophores, which cannot be recognized by neutrophils that produce lipocalin-2 [55]. In addition, the host’s neutrophil-derived protein, calprotein, is responsible for the sequestration of zinc and manganese and restriction of the growth of uropathogens [53]. The pathogenesis and defenses of the host with regard to acute cystitis and potential treatment strategies are summarized in Table 4.

Recently, emerging evidence has suggested that some biomarkers related to host immune responses may play important roles in predicting rUTIs in patients with a first UTI [56]. Potential biomarkers included urinary IL-8, urinary neutrophil gelatinase-associated lipocalin, serum vitamin D, and serum colony stimulating factors [56]. However, further large, prospective studies are necessary to verify those biomarkers as predictors of rUTI and identify their tole in the prevention and treatment of rUTIs.

### 3.3. Urinary Microbiome in Acute Cystitis

The discovery of microbial communities in the lower urinary tract using recent advanced technologies, such as culture-independent DNA sequencing, has changed the perception that urine is sterile. More than 100 species from more than 50 genera were observed in the human urogenital tract [57]. Subsequently, studies have suggested that the imbalance of the resident urinary microbial community may potentially be associated with UTIs [58]. The presence of different bacterial species in the male and female urinary microbiomes suggests that gender-specific factors may play a role in the colonization and maintenance of these microbial communities [59,60]. *Shigella, Lactobacillus, Enterococcus, Gardnerella, Prevotella, Sneathia, Escherichia*, and *Streptococcus* are observed to be prevalent in the female urobiome [61,62], while *Corynebacterium* has been identified as the dominant genus in the male urobiome [59]. The characteristics of the predominant urobiome are summarized in Table 5.

*Lactobacillus*, the dominant genus in the healthy female urinary and vaginal microbiome, enables the maintenance of a lower pH level due to the presence of lactic acid and hydrogen peroxide and inhibits the growth of uropathogens, degrades mycotoxins, and becomes a probiotic [63,64,65,66,67]. An imbalance from the healthy microbiome, for example, decreased levels of *Lactobacillus*, has been reported to be associated with rUTIs and urinary incontinence [68,69]. *Gardnerella* has been reported as a factor that has the potential to cause the re-emergence of UPEC from latent intracellular reservoirs and is associated with severe kidney infections and epithelial exfoliation [70,71] Therefore, an overgrowth of *Gardnerella* is associated with an increased risk of UTI and pyelonephritis in women [72,73]. Notably, increases in *Gardnerella* species also lead to decreases in *Lactobacillus* species [74]. However, whether dysbiosis between bacterial species actually results in disease remains to be elucidated.

**Table 5 ijms-24-07055-t005:** Common urinary microbiome found in healthy women and men.

Urobiome	Predominantly Found in	Clinical Significance	Reference
*Shigella*	Women	It is commonly found in the urobiomes of both healthy men and women. *Shigella* species rarely causes UTI; however, if they does, they can be responsible for UTI during pregnancy:	[75]
*Lactobacillus*	Women	Inhibiting the attachment of uropathogens to uroepithelial cells; Antimicrobial activity against the uropathogens by inhibiting the growth of uropathogens; Protective and anti-inflamatory roles; A potential candidate as a probiotic for the prevention and treatment of UTIs.	[76,77,78,79]
*Gardnerella*	Women	An increase in the vaginal microbiome post sexual intercourse → bacterial vaginosis. *Gardnerella* enables triggering of UPEC-related rUTIs, causing bladder epithelial exfoliation, inducing apoptosis in the bladder epithelium, and causing kidney inflammation. An increase in *Gardnerella* results in a decrease in the level of *Lactobacillus* species in urine.	[70,80]
*Prevotella*	Women	It is commonly found in the urobiomes of both healthy men and women.	[58]
*Streptococcus*	Women	An increase in *Streptococcus* is linked to sexual intercourse.	[81]
*Corynebacterium*	Men	*Corynebacterium* species are parts of the normal microflora; Occurs in 0.2% of vaginal microflora samples of healthy women; Occurs in the semen of men with prostatitis; Found in the urine of patients with an overactive bladder and urinary incontinence.	[82,83]

## 4. MDR Uropathogens

Antibiotics, especially repetitive antibiotic therapy (“collateral damage”), are the main cause for the development of antibiotic resistance, and such prolonged treatments can lead to increased resistance [84,85]. Unfortunately, almost half of patients with uncomplicated UTIs receive inappropriate or suboptimal prescriptions of antibiotics, which fail to eliminate uropathogens and can lead to a higher prevalence of MDR uropathogens, an increase in recurrent UTIs, and higher healthcare costs [86,87]. In recent decades, the incidence of highly resistant Escherichia coli and other enteric bacteria has been on the rise, making up an estimated 25–54% of all uropathogens [88].

The vast majority of UPEC shows intracellular invasion within 48 h after infection and forms biofilms, which protect from environmental challenges, including antibiotics and the host immune response [89]. Uropathogens respond to antibiotic treatment by regulating stress response mechanisms, including mutation and modification of the genome, production of enzymes to defend against antibiotics, and regulation of membrane permeability to resist antibiotic stress [90,91,92]. In antibiotic stress responses, uropathogens utilize horizontal gene transfer, which is the movement of genetic information between organisms and a process for the spread of antibiotic resistance genes among bacteria to fuel uropathogen evolution. The overexpression of marA in uropathogens is a concerning finding, as it can contribute to the development of multidrug resistance and increase the difficulty of treating UTIs [93]. MarA has been shown to activate several important virulence factors in UPEC, which can enhance the ability of the pathogen to colonize and infect the urinary tract [93]. Additionally, marA is involved in the regulation of the multidrug efflux system, which can help uropathogens to resist the effects of antibiotics [94,95,96]. Other genes have also been identified as playing a role in the antibiotic resistance of uropathogens, including rapA-regulated genes, yafQ, ymgB, and yhcQ [97,98,99]. Further research is needed to fully understand the mechanisms by which these genes contribute to antibiotic resistance in uropathogens and to develop new strategies for the prevention and treatment of antibiotic-resistant UTIs.

## 5. Emerging Prevention and Treatment Options Targeting UPEC

Because of the increasing antibiotic resistance of uropathogens, alterations in the physiological gut microbiota during treatment, and challenges in the prevention of rUTIs, different therapeutic and preventive strategies have been developed to target the pathogenicity of uropathogens. In addition to antibiotics for the treatment of symptomatic acute cystitis and low-dose antibiotic prophylaxis for rUTIs [84], alternative treatment options can be considered as a preventive strategy for UTIs (as summarized in Figure 2), although evidence of the efficacy of these options remains limited [85,100].

### 5.1. Decrease in the Periurethral Colonization of Uropathogens

Strategies to decrease periurethral colonization by uropathogens include behavioral modifications and vaginal hormonal replacement for rUTI prevention. Behavioral modifications include sufficient hydration with adequate urine output, avoidance of habitual and post-coital delayed urination, wiping from front to back after defecation, and the development of good hygiene habits [69].

#### 5.1.1. Topical Estrogen Therapy and Vaginal Laser Therapy

Vaginal estrogen transition reduces the vaginal colonization of uropathogens and further limits rUTIs through the following means: (1) an increase in the *Lactobacillus* population, which restores the vaginal flora similar to that of a premenopausal state, with an accompanying acidifying vaginal pH level [101]; (2) induction of the expression of antimicrobial peptides, which enhances the antimicrobial capacity of the uroepithelium and restricts the replication of uropathogens [102]; and (3) an increase in the expression of intercellular junction proteins, which strengthens the attachment between uroepithelial cells, preventing uropathogens from coming into contact with deeper layers of the uroepithelium and limiting the formation of reservoirs for rUTIs [103].

Meta-analyses have shown that topical estrogen (either as creams, pessaries, or per-vaginal tablets)—not oral estrogen—is an effective prophylaxis for rUTIs in women [84,104,105,106], although vaginal estrogen is not approved by the US Food and Drug Administration for the prevention of UTIs. The administration of weekly topical doses ≥ 850 μg is associated with high efficacy [107]. However, 22–67% of women undergoing vaginal estrogen therapy were reported to have rUTIs [85,108]. Hence, additional combined strategies should be considered in future studies.

Recently, vaginal laser therapy has been used to restore vaginal mucosal thickness, lubrication, and elasticity, with good effect in menopausal women with a genitourinary syndrome of menopause (GSM), which results in vaginal dryness, pain, and UTI. The authors are aware of an ongoing study investigating fractional CO_2_ vaginal laser therapy against vaginal estrogen therapy for improvement in rUTI [109]; however, the data are not yet available.

#### 5.1.2. Prophylaxis with Probiotics

Similar to the gastrointestinal tract, the urogenital tract contains a variety of microbes that are critical for the balance of the urinary microenvironment by altering local metabolite concentrations and pH levels [110]. Through next-generation sequencing and advanced culturing techniques, urinary microorganisms have been found to differ between healthy populations and those with UTIs [111], which points to the existence of a bladder microbiome [63,111]. Beneficial urinary *Lactobacillus* species, which produce lactic acid and decrease the vaginal pH, have been shown to decrease in patients with UTIs [112]. Therefore, probiotic supplementation could theoretically enhance protective microbe colonization, compete for nutrients, inhibit uropathogen biofilms, and potentially prevent UTIs [58]. Six meta-analyses and one meta-analysis targeting children were identified [84,113,114,115,116,117]; three meta-analyses reported significant positive effects of using probiotics for rUTI prevention compared to placebo in women and children [113,114,115], while the other three did not demonstrate a significant benefit in reducing UTI recurrence with probiotics compared to placebo in women [63,84,85,105,106,107,108,109,110,111,112,113,114,115,116,117]. Among the included studies, there was a lack of standardization of dose, strain, and frequency, which may impact the quantitative analysis of the outcomes, leading to contradictory results.

In addition, most studies have found that not all *Lactobacillus* species are effective, which may lead to inconsistent results. The highest efficacy was shown with *L. rhamnosus* GR-1, *L. reuteri* B-54, *L. reuteri* RC-14, *L. casei shirota*, and *L. crispatus* CTV-05 [84,113,114,117]. Therefore, probiotics containing these five strains were recommended by the 2022 EAU guidelines for the prevention of rUTI [118].

### 5.2. Antiadhesive Treatments

Antiadhesive treatments include cranberries, D-mannose, glycosaminoglycan (GAG)-layer substituents, and vaccines targeting bacterial adhesion.

#### 5.2.1. Cranberries (*Vaccinium macrocarpon*)

Cranberries contain high concentrations of proanthocyanins (PACs) and oligo- or polymers of monomeric flavan-3-ols produced as an end product of the flavonoid biosynthetic pathway. These compounds have been found in several plants, such as cranberry, blueberry, and grape seeds [119]. The advantages of PACs against UTIs include (1) anti-adhesion against P fimbriae of UPEC adherence to uroepithelial cell receptors, which is activated when the concentration of PAC is 2.4 mg/mL [120]; (2) an inhibitory effect on the bacterial motility of UPEC, *Pseudomonas aeruginosa*, and *Proteus mirabilis* [121,122]; (3) anti-biofilm activity, which is effective when the concentration of PAC is 15 μg/mL (MBIC_50_) from cinnamon [123]; (4) attenuation of the uropathogen reservoir in the gastrointestinal tract; (5) stimulation of innate immune defense in the kidney by increasing the secretion of Tamm–Horsfall protein [124,125]; and (6) synergistic effects with antibiotics against bacteria through the repression of the intrinsic resistance mechanism and additional biofilm inhibition of the action of existing antibiotics [126,127].

A Cochrane review updated in 2012 reported that cranberries do not have significant benefits in terms of decreasing the number of UTIs, and a high dropout rate was found in the cranberry group [128]. However, recent meta-analyses and systematic reviews concluded that cranberry-containing products may protect against UTIs in certain patient populations [84,129,130,131,132], especially in women, children (2–17 years), and middle-aged adults (36–55 years) [129]. The daily recommended dosage of PACs to decrease episodes of UTIs is reported to be at least 36 mg [129]. The different outcomes across the meta-analyses can be attributed to the heterogeneity in clinical diagnosis (e.g., absence of confirmation of *E. coli* UTI) and methodology of the included studies [133]. Among the included studies, the study duration ranged from 35 days to 12 months; therefore, there is no evidence to support the efficacy of cranberries for chronic use, especially for more than one year. Due to the disagreement in scientific evidence, the 2022 EAU guidelines do not recommend cranberry products for the prevention of UTIs [118].

#### 5.2.2. D-Mannose

D-mannose is a low-molecular-weight compound and a FimH receptor analogue recognized and bound by uropathogens. It is a food-derived product with medical benefits, as well as nutritional benefits, and is capable of blocking the adherence of bacteria to uroepithelial cells mediated by FimH adhesin, which functions as the tip subunit of type 1 pili [134,135,136,137]. Mannose is orally bioavailable, rapidly absorbed, and safely secreted in the urine.

The effects of mannose were proven in vivo, which included: (1) prevention of bladder uropathogen colonization and invasion, (2) effectiveness against multidrug-resistant UPEC, (3) enhancement of the effect of antibiotics, and (4) enabling the treatment of UTIs and catheter-associated UTIs [136,137,138,139].

One meta-analysis and two systematic reviews, including four randomized controlled trials and several studies, were identified [140,141,142,143,144], which concluded that D-mannose reduced rUTI significantly in subjects, both with and without catheter and prolonged UTI-free duration. However, similar to cranberries, the dose, regime, and duration lack consistency among different clinical studies [142,143]. The other limitations were small sample sizes (less than 100 patients), non-blinded studies, and self-reported rUTIs. Overall, the 2022 EAU guidelines recommend the use of D-mannose to reduce recurrent UTI episodes; however, patients should be informed that further studies are warranted to confirm the efficacy of D-mannose in initial trials [118].

#### 5.2.3. GAG Layer Substituents

The glycosaminoglycan (GAG) layer, which is coated on the bladder epithelium, has a nonspecific antiadherence role and a nonspecific protective mechanism against UTIs. Re-establishment of the GAG layer can be achieved by two main types of GAG: non-sulfated GAG, namely hyaluronic acid (HA); and sulfated GAG, namely chondroitin sulfate (CS), heparan sulfate, heparin, dermatan sulfate, and keratan sulfate. GAG layer replenishment of the bladder epithelium, which has been effectively used in different chronic inflammatory bladder diseases, such as interstitial cystitis, overactive bladder, and radiation cystitis, is a promising strategy for the management of rUTIs [144]. Two randomized controlled trials conducted to evaluate the efficacy of HA plus CS in intravesical instillation for adult women confirmed the positive effect of reducing the frequency of rUTIs and increasing the interval before the first rUTI [145,146]. In the 2022 EAU guidelines, intravesical GAG therapy was weakly recommended in patients who failed to prevent rUTI using less invasive preventive approaches [118].

### 5.3. Alternative Antibacterial Management

#### 5.3.1. Methenamine Hippurate

Methenamine hippurate, a urinary antiseptic and non-antibiotic alternative, is hydrolyzed to formaldehyde in acidic environments, such as the distal tubules in the kidney. Formaldehyde is bactericidal and functions by denaturing bacterial proteins and nucleic acids. Methenamine hippurate does not have the potential to cause the same antibiotic resistance as daily low-dose antibiotics in the prevention of rUTI and may be considered as a first-line therapy in the future.

A 2012 evidence-based Cochrane review concluded that methenamine hippurate has the benefit of preventing UTIs [147]. In contrast, another meta-analysis conducted in 2021 based on six RCTs comparing intervention to both control/no treatment and to any antibiotics and only considering women in the community suggested that no statistically significant outcome was found between methenamine hippurate and any comparators [148]. More recently, two RCTs, both comparing methenamine hippurate with the current standard care of daily low-dose antibiotics for 6–12 months, showed that methenamine hippurate is not inferior to low-dose antibiotics in preventing rUTI for 12 months after the initiation of prophylaxis [149,150]. Because of these controversial results, the 2022 EAU does not recommend the use of methenamine hippurate [118].

#### 5.3.2. Immunomodulation Therapy

Hyperactivation of IL-1β triggered by UPEC infection of the uroepithelium has been prominently identified in the pathogenicity of acute cystitis [47]. Indeed, IL-1β, a potential proinflammatory cytokine, amplifies the innate immune response in chronic infection/inflammatory models, such as tuberculosis, cystic fibrosis, and inflammatory bowel disease [151,152,153,154]. IL-1β and its receptor may be potential treatment targets for acute cystitis; therefore, the recombinant human IL-1 receptor antagonist (IL-1RA), which is an inhibitor of IL-1β hyperactivation, was developed.

To demonstrate the anti-inflammatory effects of IL-1RA in several diseases, including cystic fibrosis and COVID-19 [155,156,157], IL-1RA was studied in mice infected with UPEC [156], in which IL-1RA therapy accelerated bacterial clearance. Additionally, a similar effect was shown to be comparable to cefotaxime treatment. Based on the promising results of innate immunomodulation therapy in a mouse model of acute cystitis, further clinical trials of IL-1RA therapy for acute cystitis are required in the future.

#### 5.3.3. Vaccine Prophylaxis

Antibacterial vaccination strategies have been recently implemented to boost adaptive immunity due to their ability to prevent infection by highly virulent organisms.

##### Vaccines Targeting Bacterial Adhesion

Vaccines targeting bacterial adhesion have been developed in two domains: (1) bacterial adherence, which is related to pili or fimbriae, for example, a vaccine with purified wild-type Dr fimbriae can induce a humoral immunogenic response and further produce antibiotics against UTI but could cannot decrease the colonization in the bladder and kidney [158,159]; (2) adhesive proteins, namely adhesion–chaperone complexes, are effective in blocking the interactions between uropathogens and hosts and further protect the host against UTIs [160,161,162]. Gary et al. [163] conducted a phase I, open-label, dose escalation study to evaluate the efficacy of a FimC-FimH vaccine in 67 healthy women with and without histories of rUTI, and the results showed that all subjects tolerated the vaccine well, with greater than 150-fold increases in antibodies against the N-terminal region of FimH in the women with histories of rUTI; a further double-blind, randomized, placebocontrolled phase 2 study is currently being conducted.

##### Vaccines Containing Bacterial Extracts

Uro-Vaxom (OM-89), which contains lyophilized UPEC lysates, is administered as a daily oral tablet to prevent rUTIs by stimulating the host immune system to produce cytokines and antibodies [84,164,165,166]. The EAU guidelines recommend the prophylactic use of OM-89 because several meta-analyses have demonstrated its effectiveness in preventing rUTIs compared to a placebo at the 6-month follow-up [84,164,165,166], in which OM-89 resulted in a 39% reduction in UTIs, with minimal adverse effects compared to the placebo. Therefore, the 2022 EAU guidelines strongly recommend the use of OM-89 in female patients with rUTIs [118].

##### Vaccines Targeting Bacterial Toxins and Proteases

Vaccines targeting bacterial toxins have been developed, including vaccines with (1) toxin HlyA, which reduced renal scarring but did not protect against the colonization of UPEC in the kidneys [167]; (2) HpmA, the pore-forming secreted hemolysin of *P. mirabilis,* which did not reduce bacterial colonization [168]; and (3) proteus toxic agglutinin (Pta), a surface-associated, calcium-dependent alkaline protease that exhibits time- and dose-dependent cytotoxicity with cultured epithelial cells, which reduces kidney bacterial colonization, although it does not provide protection against bladder colonization [168].

##### Vaccines Targeting Siderophores

Vaccines targeting bacterial nutrient deprivation have been developed based on different iron acquisition systems of uropathogens, including (1) ferric yersiniabactin uptake receptor (FyuA), which showed protective effects against pyelonephritis in a murine model [169]; (2) heme acquisition protein (Hma), which provided significant protection from UPEC strain 536 bladder colonization [170]; (3) iron suptake transport aerobactin receptor (IutA), which protected the bladder from experimental challenge with UPEC by generating significant induction of antigen-specific IgA secretion that is detectable in the urine of immunized animals [170]; and (4) siderophore receptor iron-responsive element A (IreA), which reduced colonization by UPEC strain 536 in the kidney [170].

## 6. Conclusions

This review highlights the importance of the pathogenicity of uropathogens causing acute cystitis, the level of evidence for currently available treatments, and potential treatment and preventive strategies derived from pathogenic mechanisms. Various strategies for UTI management and rUTI prevention other than antibiotics present both challenges and opportunities. The targeting of multiple pathogenic mechanisms is expected to be a future trend in the management of UTIs to reduce the occurrence of rUTIs, reduce the use of antibiotics, and further decrease antibiotic resistance.

## Figures and Tables

**Figure 1 ijms-24-07055-f001:**
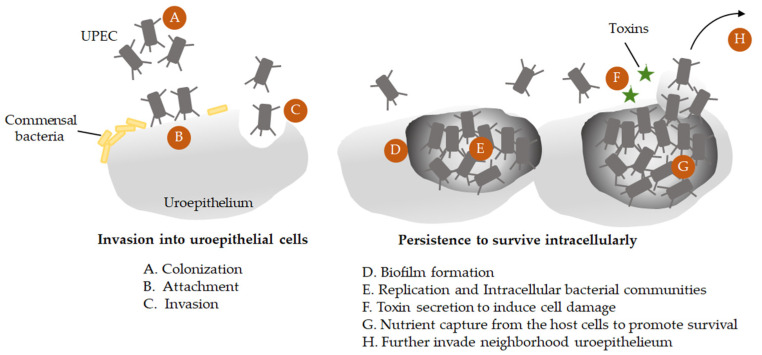
The invasive process of uropathogenic *Escherichia coli* into the uroepithelium.

**Figure 2 ijms-24-07055-f002:**
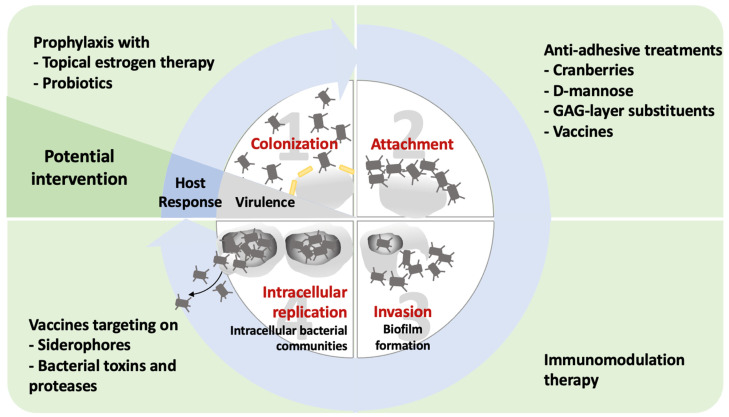
Potential antivirulence intervention for uropathogenic *Escherichia coli*. Schematic diagram illustrating alternative treatment options, including prophylactic strategies, antiadhesive treatments, immunomodulation therapy, and potential vaccines, that target the virulence of uropathogenic *Escherichia coli*.

**Table 1 ijms-24-07055-t001:** Common strains of UPEC used in UTI studies with different serogroups.

Strain	Serotype	Iron Uptake System	Source	Characteristics	Reference
536	O6:K15:H31	Yersiniabactin, enterobactin, salmochelin, and hemin uptake system	From a patient with acute pyelonephritis	Increased hemolytic activity (stronger than CFT073)	[15,16]
CFT073	O6:K2:H1	Enterobactin, salmochelin, aerobactin, hemin uptake system, and iron/manganese transport	Blood and urine of hospitalized women with acute pyelonephritis	Increased hemolytic activity. Encoding a gene cluster (*c1931-c1936*), that can produce F9 fimbria	[17,18]
UTI89	O18:K1:H17	Yersiniabactin, enterobactin, salmochelin, hemin uptake system, and iron/manganese transport	Urine of a patient with acute cystitis	Decreased endogenous production of ROS ^1^ Multiple genes related to biofilm formation were identified	[13,19]
F11	O139:H38	FetMP system	From a patient with acute uncomplicated cystitis and bacteriuria	Able to use both oxidation states of iron efficiently under conditions of ever-changing Fe(II)/Fe(III) ratios, aiding in colonization of the urinary tract	[20]
ABU83972	*Ont:*K5	Yersiniabactin, enterobactin, salmochelin, aerobactin, hemin uptake system, and iron/manganese transport	Urine of a female patient with asymptomatic bacteriuria	Rapid growth in urine	[20]

^1^ ROS = reactive oxygen species.

**Table 2 ijms-24-07055-t002:** Various adhesive fimbriae, toxins, and autotransporters contributing to virulence in UPEC.

Virulence Factors	Type of Virulence Factors	Receptor	Gene	Characteristics	Reference
Adhesins	Type I fimbriae	Monomannose moiety of the tetraspanin molecule uroplakin 1a (UP1a)	*fim* B, *fim* E, *fim* H, and *Pil*	. Mannose-sensitive . Responsible for colonization, invasion, and persistence of uroepithelial cells . The most frequently expressed virulence factor, accounting for 80 to 100% of UPEC . Less prevalent in pyelonephritogenic UPEC but prevalent in urinary catheter-related bacteria	[27,28]
	P fimbriae	Globoseries glycosphingolipid receptors (Gal-Gal)	*PapG* and *pap*GAP	. Mannose-insensitive . Able to bind to glycosphingolipids of the kidney epithelium . A super-virulence factor that transits the response from asymptomatic bacteriuria to acute pyelonephritis in the host . Associated with the severity of UTI	[29,30]
	Dr adhesins	Type 4 collagen and Dr blood group antigen	*AfaE1–5, AfaF*, and *Drb* operon	. Invasion of epithelial cells of the bladder . Consists of both fimbrial and afimbrial adhesins	[28]
	S and F1C fimbriae	Sialyl-α-2-3 galactoside	*Sfa/fac*	. Associated with pyelonephritis and cystitis . Expressed by approximately 14% of UPEC	[31,32]
Toxins	α-hemolysin (HlyA)	CD11a/CD18 (LFA-1) and glycophorin	*hly*	. Isolated from more than 70% of patients with pyelonephritis . Isolated from 31–48% of patients with acute cystitis . Associated with renal complications, including permanent renal scarring, in up to 50% of pyelonephritis cases	[33]
	Cytotoxic necrotizing factor 1	Lu/BCAM adhesion glycoprotein	*cnf1*	. Able to modulate the activity of Rho GTPases	[34,35,36]
Autotransporters	Vacuolating autotransporter toxin (VAT)	Cytokeratin 8	*vat*	. Identified in 20–36% of UPEC . Isolated from 59–68% of patients with pyelonephritis and urosepsis	[24,37,38,39,40]
	Secreted autotransporter toxin (SAT)	Cytokeratin 8	Sat-encoding gene *(sat)*	. Isolated from 68% of patients with pyelonephritis	[40]

**Table 3 ijms-24-07055-t003:** Mechanisms of survival of UPEC within urothelial cells.

Aim of UPEC	Strategies of UPEC	Mechanism	Reference
Antagonism of uroepithelial cell apoptosis	Aerobic respiration	Cytochrome *bd* (reducing the efficiency of mitochondrial respiration) → stabilize HIF-1 → promote aerobic glycolysis	[43]
Maintenance of uropathogens’ cellular function	Nutrient acquisition	Iron: extracellular insoluble/biounavailable Fe^3+^ + siderophore → soluble/bioavailable siderophore- Fe^3+^ complex → transport into cell → intracellular iron assimilation from the siderophore–Fe^3+^ complex → intracellular iron usage and storage	[46]
Enhancement of uroepithelial cell exfoliation	Toxin secretion	Toxins → cell damage and apoptosis→ release of iron and nutrients from host cells	[14]

**Table 4 ijms-24-07055-t004:** The pathogenesis and defenses of the host with regard to acute cystitis and potential treatment strategies.

Steps of UPEC Pathogenesis	Normal Defenses of Hosts
Colonization of uropathogens in the urethra, periurethra, and vagina	Normal commensal flora form a barrier against the colonization of uropathogens The natural flow of urine with high osmolality and low pH and containing soluble IgA immunoglobulins or Tamm–Horsfall protein (uromodulin)
Fim-H-mediated adherence to epithelial cells of the bladder	Inflammatory responses in the lower urinary tract Induction of bacterial clearance by purging the infected uroepithelial cells, which undergo rapid cell death and exfoliation [47]
Biofilm elaboration and intracellular replication	Urothelial exfoliation Active expulsion of UPEC by increased intracellular cyclic AMP in urothelial cells [48,49]
Formation of bladder intracellular bacterial communities for invasion and replication	Autophagy for the elimination of uropathogens from infected host cells
Siderophores for iron acquisition to maintain survival	Host neutrophils produce lipocalin-2 capable of binding enterobactin siderophore against enterobactin-mediated iron uptake by UPEC
Zinc acquisition to maintaining survival	Host neutrophil-derived protein calprotein enables the sequestration of zinc

## Data Availability

Not applicable.

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
