# Peer review of "Emerging Non-Antibiotic Options Targeting Uropathogenic Mechanisms for Recurrent Uncomplicated Urinary Tract Infection"

_ijms, 2023, doi:10.3390/ijms24087055_

Round 1
Reviewer 1 Report
This is an interesting review on an actual medical topic. The review is well structured. Given the different chapters, the article might benefit from a 'table of contents' at the beginning. The text is well written and easily readable.
Some remarks:
-line 76 to 78: what is meant by "which is not pathogenic except in immunosuppressed individuals" ? the fact that commensal microorganisms reside somewhere in the body is by definition not pathogenic in itself, so why would it be pathogenic in the immunosuppressed? and this statement does not seem to be discussed in ref [11]. Can this be clarified?
-line 122: the title indicated as 3.1 is not very clear; wouldn't it be better to say: "The adaptive evolution of UPEC in UTI to thrive within urothelial cells" ?
The statement in line 124 and line 125 is also unclear: what is meant by "...the virulence factors of UPEC are borne by and contribute to them" ? can this be explained more clearly?
-line 131: "Tamm-Horsfall proteins" in stead of "Tamm-Horsifall proteins"
-line 145: the ";" after [46] should be a ","
-line 156: rather "toxin secretion" than "toxins secretion" (or: "secretion of toxins")
-line 204: rather "becomes a probiotic" than "becomes a probiotics"
-line 248: rather "Emerging prevention and treatment options targeting UPEC"
-line 463: rather "...will be the future trend of managing UTI to reduce..."
Author Response
Dear reviewer:
We are very grateful to your comments for this manuscript. According with your advice, we amended the relevant part in manuscript. Our point-by-point responses are presented below.
Reviewer 1.
Point 1:
This is an interesting review on an actual medical topic. The review is well structured. Given the different chapters, the article might benefit from a 'table of contents' at the beginning. The text is well written and easily readable.
Response 1: Thank you for your constructive suggestion. We have added on a table of content at the beginning of this manuscript
Some remarks:
Point 2: -line 76 to 78: what is meant by "which is not pathogenic except in immunosuppressed individuals" ? the fact that commensal microorganisms reside somewhere in the body is by definition not pathogenic in itself, so why would it be pathogenic in the immunosuppressed? and this statement does not seem to be discussed in ref [11]. Can this be clarified?
Response 2: Thank you for your comment. We have deleted the this sentence and reference 11 in the revised munuscript. As the reviewer indicated, commensal bacteria, serve to maintain microbial stability and colonization resistance by preventing overgrowth or domination with more pathogenic bacteria, through interactions within the microbial community and with the host. In patients with immune suppression, disruption of the microbiota gives rise to perturbations by pathogenic species, leading to increased bacterial translocation and susceptibility to systemic infection [1].
Reference 1: Taur Y, Pamer EG. The intestinal microbiota and susceptibility to infection in immunocompromised patients. Curr Opin Infect Dis. 2013, 26, 332-7.
Point 3: -line 122: the title indicated as 3.1 is not very clear; wouldn't it be better to say: "The adaptive evolution of UPEC in UTI to thrive within urothelial cells" ?
Response 3: Thank you for your comment. We have revised the sentence according to the reviewer’s suggestion. Thank you. (Page 5, Line 157)
Point 4: The statement in line 124 and line 125 is also unclear: what is meant by "...the virulence factors of UPEC are borne by and contribute to them" ? can this be explained more clearly?
Response 4: Thank you for your comment. We have deleted this sentence in the revised manuscript. Thank you.
Point 5: line 131: "Tamm-Horsfall proteins" in stead of "Tamm-Horsifall proteins"
Response 5: Thank you for your comment. We haved corrected it. (Page 6, Line 167)
Point 6: line 145: the ";" after [46] should be a ","
Response 6: Thank you for your comment. We haved corrected it. (Page 6, Line 182)
Point 7: line 156: rather "toxin secretion" than "toxins secretion" (or: "secretion of toxins")
Response 7: Thank you for your comment. We haved corrected it. (Page 6, Line 193)
Point 8:line 204: rather "becomes a probiotic" than "becomes a probiotics"
Response 8: Thank you for your comment. We haved corrected it. (Page 8, Line 252)
Point 9: line 248: rather "Emerging prevention and treatment options targeting UPEC"
Response 9: Thank you for your comment. We haved corrected it. (Page 10, Line 316)
Point 10: line 463: rather "...will be the future trend of managing UTI to reduce..."
Response 10: Thank you for your comment. We haved revised it in the revised manuscript.
(Page 15, Line 554)
Reviewer 2 Report
The review article by Chen Y. et al. have systematically studied the importance of the pathogenicity of uropathogens causing acute cystitis, the level of evidence for currently available treatments, and potential treatment and preventive strategies derived from pathogenic mechanisms for rUTI.
The manuscript is well written with clear description. However, some concerns should be addressed.
1. The title is so big and not aiming the content of review article. Please make this tile more focus for the researchers.
2. As urine or serum biomarkers for predicting recurrence in patients with a first UTI is important and clinically useful. Authors need to talk more about this.
3. There are several reviews articles those have already been talked about the current available treatments for rUTI, so how authors make this review article different then others.
4. The interval time of references in the introduction is too long and contains a maximum of older references, so it is suggested to quote the literatures in the last three to five years.
5. Authors need to write the discussion part in a descriptive way with explanation.
6. The language expression of the manuscript needs to be further simplified and polished.
Author Response
Reviewer 2.
The review article by Chen Y. et al. have systematically studied the importance of the pathogenicity of uropathogens causing acute cystitis, the level of evidence for currently available treatments, and potential treatment and preventive strategies derived from pathogenic mechanisms for rUTI.
The manuscript is well written with clear description. However, some concerns should be addressed.
Response: We thank the Reviewer for the affirmative views on this manuscript, and appreciate the opportunity to further improve on our manuscript.
Point 1: The title is so big and not aiming the content of review article. Please make this tile more focus for the researchers.
Response 1: Thank you for your comment. We have revised the title to “Emerging Non-Antibiotic Options Targeting on Uropathogenic Mechanisms for Recurrent Uncomplicated Urinary Tract Infection” in the revised manuscript.
Point 2 : As urine or serum biomarkers for predicting recurrence in patients with a first UTI is important and clinically useful. Authors need to talk more about this.
Response 2: Thank you for your comment. We have added “ Recently, emerging evidence suggest that some biomarkers related to host immune responses may play important roles in predicting rUTI in patients with a first UTI [Ref 1]. Potential biomarkers included urinary IL-8, urinary neutrophil gelatinase-associated lipocalin, serum vitamin D, serum colony stimulating factors [Ref 1]; However, further large, prospective studies are necessary to verify those biomarkers as a predictor of rUTI and identify their tole in the prevention and treatment of rUTI. “ in the revised manuscript. (Page 7, Line 227-233)
Reference 1: Sorić Hosman I, Cvitković Roić A, Lamot L. A Systematic Review of the (Un)known Host Immune Response Biomarkers for Predicting Recurrence of Urinary Tract Infection. Front Med (Lausanne). 2022, 4, 931717.
Point 3 : There are several reviews articles those have already been talked about the current available treatments for rUTI, so how authors make this review article different then others.
Response 3: Thank you for this critique. We have re-considered this issue. In fact, this manuscript provides updating non-antibiotc options for targeting on the pathogenicity of uropathogens, which may contribute to the future trends of researches and treatment strategies to limit rUTI. For this purpose, we have added on a new Figure 2 to illustrate our aim in this review article.
Point 4 : The interval time of references in the introduction is too long and contains a maximum of older references, so it is suggested to quote the literatures in the last three to five years.
Response 4: Thank you for your comment. We have updated the references in the revised manuscript.
Point 5 : Authors need to write the discussion part in a descriptive way with explanation.
Response 5: Thank you for your comment. We have revised our manuscript by this request. (Page 9, Line 266-279)
Point 6 : The language expression of the manuscript needs to be further simplified and polished.
Response 6: Thank you for this comment. Some sections have been simplified in the revised manuscript. (Page 7-8: section 3.3. Urinary microbiome in acute cystitis; page 11: section 5.1.1. Topical estrogen therapy and vaginal laser therapy; page 11: section 5.1.2 Prophylaxis with probiotics; page 13: section 5.3.1 Methenamine hippurate)